# Monocyte Distribution Width, Neutrophil-to-Lymphocyte Ratio, and Platelet-to-Lymphocyte Ratio Improves Early Prediction for Sepsis at the Emergency

**DOI:** 10.3390/jpm11080732

**Published:** 2021-07-28

**Authors:** Sen-Kuang Hou, Hui-An Lin, Shao-Chun Chen, Chiou-Feng Lin, Sheng-Feng Lin

**Affiliations:** 1Department of Emergency Medicine, Taipei Medical University Hospital, Taipei 110, Taiwan; 992001@h.tmu.edu.tw (S.-K.H.); sevenoking219@gmail.com (H.-A.L.); 2Department of Emergency Medicine, School of Medicine, College of Medicine, Taipei Medical University, Taipei 110, Taiwan; shaochun36@gmail.com; 3Graduate Institute of Injury Prevention and Control, College of Public Health, Taipei Medical University, Taipei 110, Taiwan; 4Department of Microbiology and Immunology, School of Medicine, College of Medicine, Taipei Medical University, Taipei 110, Taiwan; cflin2014@tmu.edu.tw; 5School of Public Health, College of Public Health, Taipei Medical University, Taipei 110, Taiwan

**Keywords:** monocyte distribution width (MDW), sepsis, systemic inflammatory response syndrome (SIRS), Sequential Organ Failure Assessment (SOFA)

## Abstract

(1) Background: Sepsis is a life-threatening condition, and most patients with sepsis first present to the emergency department (ED) where early identification of sepsis is challenging due to the unavailability of an effective diagnostic model. (2) Methods: In this retrospective study, patients aged ≥20 years who presented to the ED of an academic hospital with systemic inflammatory response syndrome (SIRS) were included. The SIRS, sequential organ failure assessment (SOFA), and quick SOFA (qSOFA) scores were obtained for all patients. Routine complete blood cell testing in conjugation with the examination of new inflammatory biomarkers, namely monocyte distribution width (MDW), neutrophil-to-lymphocyte ratio (NLR), and platelet-to-lymphocyte ratio (PLR), was performed at the ED. Propensity score matching was performed between patients with and without sepsis. Logistic regression was used for constructing models for early sepsis prediction. (3) Results: We included 296 patients with sepsis and 1184 without sepsis. A SIRS score of >2, a SOFA score of >2, and a qSOFA score of >1 showed low sensitivity, moderate specificity, and limited diagnostic accuracy for predicting early sepsis infection (c-statistics of 0.660, 0.576, and 0.536, respectively). MDW > 20, PLR > 9, and PLR > 210 showed higher sensitivity and moderate specificity. When we combined these biomarkers and scoring systems, we observed a significant improvement in diagnostic performance (c-statistics of 0.796 for a SIRS score of >2, 0.761 for a SOFA score of >2, and 0.757 for a qSOFA score of >1); (4) Conclusions: The new biomarkers MDW, NLR, and PLR can be used for the early detection of sepsis in the current sepsis scoring systems.

## 1. Introduction

Sepsis, a life-threatening condition, is substantially associated with high morbidity and mortality in health care systems worldwide [1,2,3,4,5,6]. Early diagnosis of sepsis, along with prompt initiation of treatment, can result in favorable outcomes [7,8,9]. Physicians in the emergency department (ED) who first encounter a patient in health settings should detect any sign of sepsis progression [10]. Currently, the recommended approach to sepsis is based on two definitions: (1) as per Sepsis-2 criteria published in 1992, sepsis is defined as documented infection in conjugation with a systemic inflammatory response syndrome (SIRS) score of ≥2 points [11]; and (2) as per Sepsis-3 criteria published in 2016, sepsis is defined as a quick sequential organ failure assessment (qSOFA) score of ≥2 [12] followed by an acute change of ≥2 points in the SOFA score [13] due to the infection [14]. The complexity of the definition of sepsis and the absence of an effective biomarker impede the early diagnosis of sepsis in the ED.

To date, numerous biomarkers such as interleukin-6 (IL-6), C-reactive protein (CRP), and procalcitonin (PCT), have been used for the early detection of sepsis [9,15,16,17]. However, in real-world clinical practice, the advanced biomarkers are still limited. IL-6 is not readily available in ED. Despite more specific for bacterial infection [18,19,20], PCT is not available in every first-level hospital, and its cost, around four times higher than CRP, is another concern [20,21,22]. Although CRP is widely available, the turnaround time (TAT) for CRP is longer than a routine CBC report (including MDW). Emergency physicians may order these tests based on the results of routine complete blood cell count (CBC) analysis performed using a hematology analyzer. Under most conditions, the TAT for clinical chemistry instrument is longer than TAT for hematology analyzer [23,24,25]. If a point-of-care testing of CRP was not available, this may cause a delay of early antibiotic use.

Monocyte distribution width (MDW) is a new biomarker analyzed in conjugation with routine CBC in a hematology analyzer [9,15,26]. In the early stage of sepsis, the proliferation of white blood cells (WBCs), namely monocytes and neutrophils, acts as the host’s first immune response against pathogens [27,28]. Leukocytosis was observed to be associated with increased volume and heterogeneity in terms of MDW [9]. In addition, other biomarkers examined in a hematology analyzer, including neutrophil-to-lymphocyte ratio (NLR) [29,30,31] and platelet-to-lymphocyte ratio (PLR) [31,32,33], are useful for the early diagnosis of sepsis. This study investigated whether MDW, NLR, and PLR, in conjugation with sepsis scoring systems, namely the SIRS, SOFA, and qSOFA, can aid in the early identification of sepsis during the first ED visit.

## 2. Materials and Methods

### 2.1. The Study Design and Data Collection

This retrospective observational study, based on a prospective registry system, was conducted in Taipei Medical University Hospital, a major tertiary referral hospital with 750 beds in Taipei, Taiwan. Since 1 January 2020, all patients visiting our ED were registered in this prospective registry system. This registry system included information regarding patients’ age, sex, vital signs, triaging level, and Glasgow coma scale (GCS) score on arrival to ED; laboratory testing data at ED; disease codes based on the International Statistical Classification of Diseases, Tenth Revision, Clinical Modification (ICD-10-CM); past history of medical comorbidities (including hypertension, diabetes mellitus, coronary artery disease, cerebrovascular disease, end-stage renal disease, pulmonary disease, and malignant disease); and disposition of ED patients (discharge or hospitalization to the general ward or intensive care unit). This study was approved by the Joint Institutional Review Board of Taipei Medical University (approval number: N201904066). The requirement of informed consent was waived by the aforementioned review board because anonymous and deidentified information was used in this study.

### 2.2. Participants

Patients who were registered in the prospective registry system of ED between 1 January 2020 and 30 November 2020 were enrolled in this study. Inclusion criteria were as follows: (1) patients presenting to our ED with infectious diseases (medical record confirmed by the authors H-A Lin and S-K Hou), (2) examination of patients by ED physicians, and (3) completion of laboratory tests within 2 h after arrival to the ED. Exclusion criteria were as follows: (1) age < 20 years, (2) no definite diagnosis of infectious disease after the quality review of our data (conducted by the authors H-A Lin and S-K Hou), and (3) patients with no laboratory tests by using a hematology analyzer at the ED.

### 2.3. Outcome Measures

The primary outcome was the early diagnosis of sepsis according to Sepsis-2 criteria. Patients with sepsis were defined as those having a documented infection and a SIRS score of ≥2 points based on the following parameters: (1) a body temperature of <36 °C or >38 °C, (2) a heart rate of >90/min, (3) a respiratory rate of >20/min or PaCO2 of <32 mmHg, and (4) a WBC count of <4000 or >12,000 cells/mm^3^ (or with 10% bands). Our patients were categorized into 2 groups: sepsis and nonsepsis. In addition, all patients were scored using the qSOFA and SOFA scoring systems based on our registry data. As a secondary outcome, we examined early mortality within 3 days due to septic shock.

### 2.4. New Biomarker Measurement

Since 1 January 2020, MDW has been measured as a new biomarker for inflammation in Taipei Medical University Hospital. MDW was analyzed using a Beckman Coulter UniCel DxH 900 analyzer (Beckman Coulter Taiwan, Taiwan Branch), which is a quantitative, multiparametric, automated hematology analyzer. On this analyzer, monocytes were identified on the basis of individual cell volume, high-frequency conductivity, and laser-light scatter. MDW was calculated as the standard deviation of a set of monocyte cell volume values. In addition, NLR and PLR were calculated by dividing neutrophil and platelet counts by the lymphocyte count, respectively.

### 2.5. Statistical Analysis

Continuous and discrete variables are presented as the mean ± standard deviation and proportion (%), respectively. A 1:4 propensity score (PS) matching was performed between the sepsis and nonsepsis groups by matching age, sex, GCS score, triage level, and medical comorbidities. A standardized mean difference (SMD) of less than ±0.1 between the 2 groups was considered to be significant. Univariate and multivariate logistic regression models were employed to obtain the odds ratios (ORs) of predictors; sepsis versus nonsepsis (according to Sepsis-2 criteria) was used as the dependent variable, and the score of each sepsis scoring system (SIRS, SOFA, and qSOFA) and biomarkers (WBC, MDW, NLR, and PLR) were used as independent variables. To perform sensitivity analysis, we used 2 multivariate models: without and with implementing CRP, an inflammatory biomarker. The receiver operating characteristic (ROC) curve was plotted to estimate the c-statistics. Optimal cutoff points were obtained by calculating Youden’s index (the point with the maximum value of sensitivity + specificity − 1). The integrated discrimination improvement (IDI) test [34] was used to compare the c-statistics between the 2 ROC curves. The Hosmer–Lemeshow test was performed to examine the goodness of fit. All statistical analyses were conducted using SAS version 9.4 (Cary, NC, USA). Statistical significance was defined as *p* < 0.05.

## 3. Results

### 3.1. Characteristics of Enrolled Participants

A total of consecutive 19,792 patients visited the ED between January 2020 and November 2020. Of them, 8698 patients were diagnosed as having an infectious disease after the data quality review by our medical record auditing team. In accordance with “Sepsis-2” consensus criteria, patients with infectious disease were categorized into two groups: sepsis (*n* = 308) and nonsepsis (*n* = 8390; Table 1). The detailed etiologies of microorganisms in sepsis group were summarized in the supplemental file (Appendix A). Before PS matching, age, GCS score, triage level, and the prevalence of hypertension, diabetes mellitus, coronary artery disease, cerebrovascular disease, and end-stage renal disease significantly differed between the two groups. After performing 1:4 PS matching, 296 and 1184 patients were included in the sepsis and nonsepsis groups, respectively. All the aforementioned demographic characteristics and medical comorbidity prevalence were similar between the 2 groups (SMD < ±0.1). The flow diagram of the study was shown (Figure 1). Data regarding the distribution and diagnostic performance of SIRS score at each point was shown (Appendix A). From 1 January 2020, and 30 November 2020, the COVID-19 was not a pandemic disease in Taiwan according to data from the Taiwan National Infectious Disease Statistics System. None of the patients enrolled in the study had the diagnosis of COVID-19, and no indigenous case during our study period was reported.

### 3.2. Predictors for Sepsis

Table 2 presents the findings of univariate logistic regression models used for analyzing the predictive value of sepsis scoring system scores and biomarkers. The scores of all sepsis scoring systems, namely the SIRS, SOFA, and qSOFA, were found to be significant predictors of sepsis. After applying the optimal cutoff point for sepsis scoring system scores, a SIRS score of ≥3 (OR: 5.46, 95% CI: 4.11–7.26, *p* < 0.0001), a SOFA score of ≥ 3 (OR: 1.90, 95% CI: 1.46–2.46, *p* < 0.0001), and a qSOFA score of ≥ 2 (OR: 1.86, 95% CI: 1.30–2.64, *p* < 0.0006) remained as significant predictors. Among these sepsis scoring systems, a SIRS score of ≥ 3, a SOFA score of ≥ 3, and a qSOFA score of ≥ 2, exhibited the low sensitivity of 44.9%, 47.0%, and 17.6%, respectively for sepsis, and the high specificity of 87.0%, 68.2%, and 89.7%, respectively, for sepsis. In terms of laboratory tests, the biomarkers WBC, NLR, PLR, CRP, and MDW were all found to be significant predictors of sepsis. We applied the following optimal cutoff points: WBC > 12 × 103 cells/µL (OR: 3.38, 95% CI: 2.59–4.52, *p* < 0.0001), NLR > 9 (OR: 5.85, 95% CI: 4.43–7.73, *p* < 0.0001), PLR > 210 (OR: 3.12, 95% CI: 2.39–4.05, *p* < 0.0001), CRP > 3 mg/dL (OR: 4.82, 95% CI: 3.66–6.36, *p* < 0.0001), and MDW > 20 (OR: 2.77, 95% CI: 2.13–3.62, *p* < 0.0001). After applying the cutoff points, we observed moderate diagnostic accuracy with moderate levels of sensitivity and specificity (i.e., 60%–70%).

### 3.3. Multivariate Models for Predicting Sepsis

We used three types of multivariate logistic regression models (incorporating a SIRS score of ≥3 in model 1, a SOFA score of ≥ 3 in model 2, and a qSOFA score of ≥2 in model 3) for predicting sepsis (Table 3). In all the models, each sepsis scoring system, NLR > 9 U and MDW > 20 U were found to be significant predictors. These models demonstrated moderate to high diagnostic accuracy (an AUC of 0.796 in model 1, an AUC of 0.761 in model 2, and an AUC of 0.757 in model 3). After adjustment for a CRP level of >3 mg/dL in these models, NLR > 9 U still remained a significant predictor but not MDW > 20 U.

### 3.4. Sensitivity Analysis: Diagnostic Performance of Multivariate Models without and with CRP

We compared the performance of each scoring system in predicting sepsis. Model 1 without CRP and with CRP exhibited a low quality of model fit (without CRP, the Hosmer–Lemeshow test, *p* = 0.0137; with CRP, the Hosmer–Lemeshow test, *p* = 0.0182). Both models 2 and 3 without CRP and with CRP revealed a good quality of model fit. Both model 2 (IDI: 0.91, *p* = 0.1924) and model 3 (IDI: 0.70, *p* = 0.3202) showed no significant difference in diagnostic accuracy between models without and with CRP. The ROC curves for three types of multivariate logistic regression models of predicting sepsis were shown (Figure 2).

### 3.5. Secondary Outcome: Predicting Mortality for Patients with Sepsis

As the secondary outcome, we assessed whether these sepsis scoring system–based models could predict early mortality within 10 days (Table 4). In general, all these models demonstrated a high diagnostic accuracy. In multivariate models without CRP, MDW > 20 U remained a robust predictor of mortality. By contrast, MDW > 20 U did not predict mortality after adjustment for CRP. In addition, no significant difference in diagnostic performance was observed between models without CRP and with CRP. The ROC curves for three types of multivariate logistic regression models of predicting mortality were shown (Figure 3).

### 3.6. A New Simple Scoring System for Predicting Sepsis

We constructed a new scoring system model for predicting sepsis (Table 5). In this model, a qSOFA score of >1, PLR > 210, and MDW > 20 U were assigned a score of 1 and NLR > 9 U was assigned a score of 2. This model showed moderate to high diagnostic accuracy with an AUC of 0.755 (95% CI: 0.726–0.784) and a good quality of model fit (the Hosmer–Lemeshow test, *p* = 0.3020). The optimal cutoff score of ≥2 exhibited a sensitivity of 77% (95% CI: 73.0%-82.4%), a specificity of 63.9% (95% CI: 61.1%-66.6%), and an OR of 6.16 (95% CI: 4.57–8.30, *p* < 0.0001). The ROC curve for this new simple scoring system was shown (Figure 4).

## 4. Discussion

The findings of this study indicated that the new inflammatory biomarkers, namely MDW > 20, NLR > 9, and PLR > 210, in conjugation with scoring systems, increased the diagnostic accuracy for early sepsis. In addition, we developed a simple 5-point model that involved a qSOFA score of >1 along with MDW > 20, NLR > 9, and PLR > 210; this model could differentiate between patients with and without the need of early antibiotics. Since 2012, the early goal-directed therapy as per Sepsis-2 criteria had emphasized the benefit of antibiotic use in the first hour after severe sepsis or septic shock impressed [35]. Every hour of delay in antibiotic administration was significantly associated with an increased mortality [36,37,38]. Early use of proper antibiotics can be accomplished through prompt and correct diagnosis of sepsis. This new model can help physicians recognize sepsis during the patient’s first ED encounter.

At an early stage of sepsis, monocytes exhibit morphological changes and heterogeneity in size [39,40,41]. MDW can indicate morphological variations in monocytes and serve as a novel inflammatory biomarker [9,42,43]. Crouser et al. demonstrated early septic patients can be confirmed in the initial 12 h of ED arrival with the elevated MDW [9,44]. The time course of biomarkers after pathogen exposure in blood usually follows by (1) leukocytosis (such as count, morphological change of WBC), (2) inflammatory cytokines (such as IL-6, TNF-α), (3) inflammatory proteins (such as CRP, PCT), (4) poor perfusion-related markers (low pH, lactate), and (5) coagulopathy (elevated D-dimers, fibrin). On average, the peak concentration of PCT was around 18 h and around 24 to 48 h for CRP. Increased PCT and CRP sends an alarm to physicians that patients should be in a later phase of sepsis compared to increased MDW. Biomarker measurement using a hematology analyzer has many advantages: (1) no requirement of additional blood collection for analysis in a clinical chemistry analyzer, (2) no time delay in obtaining the biomarker results along with routine CBC data, and (3) no large learning threshold for the interpretation of data. Moreover, recent studies have employed MDW to evaluate the severity of patients with coronavirus disease 2019 [29,45,46].

The cutoff values of the MDW were heterogenous among various studies, in which the reference interval for MDW was reported to be from 19 to 25 [9,26,42,43,47,48]. Woo et al. indicated that the cutoff value of MDW of >19.8 demonstrated optimized sensitivity and specificity for screening patients with sepsis in the ED [26]. Crouser et al. revealed that an MDW of >20.5 differentiated patients with sepsis from those without sepsis in the ED setting [9]. Furthermore, Piva et al. showed that a MDW of >24.6 predicted sepsis in ICU patients according to Sepsis-3 criteria [48]. The cutoff value of >20.0 used for MDW in this study is consistent with that used in previous studies in the ED. The heterogeneity of the cutoff values of MDW might be relevant to the clinical stages of patients with sepsis. Compared with ICU patients, most patients admitted to the ED had the early stage of sepsis. This finding can explain the higher cutoff value of MDW in studies conducted in ICU patients than in those performed in ED patients.

MDW outweighs other biomarkers examined using a hematology analyzer for early sepsis diagnosis in the ED, including the mean neutrophil volume, neutrophil distribution width, and mean monocyte volume [9]. Compared with MDW, these biomarkers exhibited delayed variations in response to sepsis [9]. Crouser et al. reported that the prediction performance for sepsis was more convincing when they used a combination of both MDW and WBC compared with MDW alone [9]. Besides, previous studies have found that NLR and PLR were significantly associated with the prediction and prognosis of sepsis [30,31,49,50,51]. Diordjevic et al. even noticed that level of NLR and PLR correlated with the results of blood culture [31]. The cutoff points of NLR [31,52,53,54,55,56,57] and PLR [31,32,56] varied among different studies. This variation could result from differences in the sepsis causes, the timeframe for examination, study participants, and test methods [30]. Our cutoff values are consistent with the reference intervals of NLR (ranging from 4.06 to 23.80) [31,52,53,54,55,56,57] and PLR (ranging from 200 to 240) [31,32,56] indicated in previous studies. Our findings indicated that NLR or PLR can be applied for early sepsis diagnosis.

We attempted to improve the diagnostic performance for early sepsis recognition by combining biomarkers examined using a hematology analyzer and sepsis scoring systems. We observed that all sepsis scoring systems had a high specificity but low sensitivity, and biomarkers examined using a hematology analyzer had both moderate sensitivity and specificity (Table 2). The biomarkers MDW, NLR, and PLR overcame the limitation of the low sensitivity of sepsis scoring systems (Table 3). Multivariate models based on SIRS, SOFA, and qSOFA scores showed similar diagnostic performance for early sepsis diagnosis. Compared with the conventional biomarker CRP, our multivariate analysis performed using models with and without the inclusion of CRP showed similar diagnostic accuracy (the IDI test for c-statistics showed no significant difference, Table 3). This result suggested that the diagnostic performance of the new biomarkers was comparable to that of CRP.

To simplify the complexity of early sepsis recognition in the ED, we developed a 5-point scoring system (Table 5) by combining qSOFA scores with MDW, NLR, and PLR. Our selection of the qSOFA score (rather than SIRS and SOFA scores) is reasonable because of the following reasons: (1) the qSOFA score assessed mental status, respiratory rate, and systolic blood pressure [12]; thus, the qSOFA score is more convenient than SIRS and SOFA scores; (2) the diagnostic performance among SIRS, SOFA, and qSOFA scores was similar in multivariate models; and (3) a meta-analysis comparing clinical assessment tools for sepsis indicated that the SIRS score had higher sensitivity, whereas the qSOFA score had higher specificity [58]. Because our models focused on increasing specificity by using a sepsis scoring system, the qSOFA score was considered more optimal than the other two scores. Finally, the high specificity of the qSOFA score observed in our study is compatible with that indicated in a previous study [58,59,60].

We performed a separate secondary outcome to determine the prediction performance of these models for early mortality within 3 days. We observed that all these multivariate models combining sepsis scoring systems with biomarkers (Table 4) had high diagnostic performance. Of them, the model employing the SOFA score had the highest accuracy. We considered that the SOFA score offers more prognostic information than the other 2 scores, and this result is consistent with that reported by Raith [61] who assessed SIRS, SOFA, and qSOFA scores. In addition, the qSOFA score was more strongly correlated with unfavorable sepsis outcomes compared with the SIRS score. Moreover, clinical practice in the ED emphasizes early sepsis recognition more than prognostic information; therefore, the qSOFA score can be more optimal. In addition, Sepsis-2 criteria–based diagnosis is more applicable in the ED than Sepsis-3 criteria-based diagnosis [62,63,64]. We adopted Sepsis-2 definition to fulfill the real-world ED setting.

## 5. Conclusions

For clinical use, we developed a tool that can diagnose sepsis in the early stages. In the present study, this new scoring system developed for predicting sepsis showed an AUC of 0.757 when a qSOFA score of >1, a NLR of >9 U, a PLR of >210, and a MDW of >20 U were employed. With a score of ≥2, this new scoring system showed good sensitivity and specificity for early identification of sepsis in the ED.

## Figures and Tables

**Figure 1 jpm-11-00732-f001:**
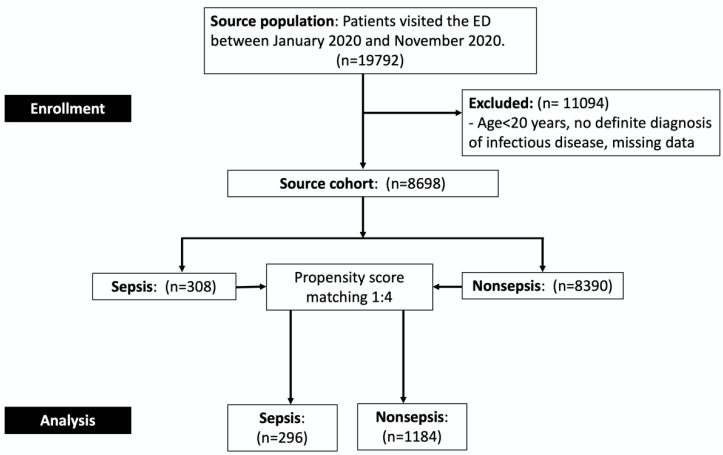
Flow diagram of the study.

**Figure 2 jpm-11-00732-f002:**
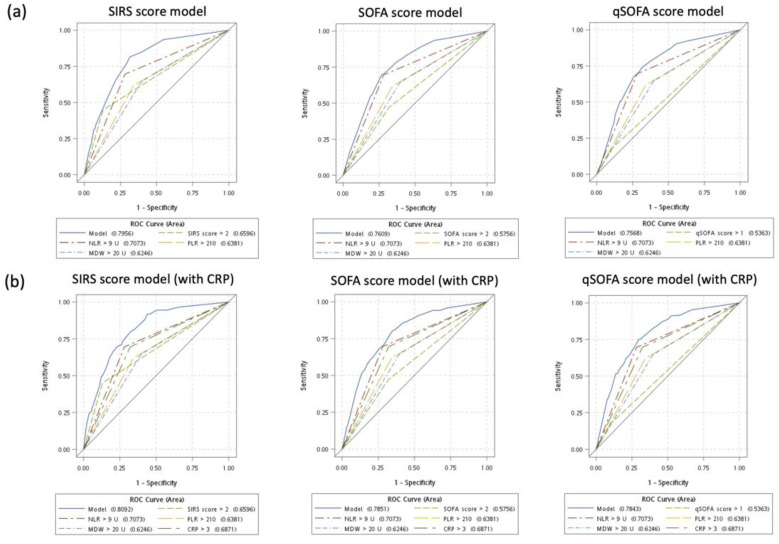
Receiver of operating characteristic (ROC) curves of predicting sepsis for models based on SIRS score, SOFA score, and qSOFA score. (**a**) Models without CRP. (**b**) Models with CRP.

**Figure 3 jpm-11-00732-f003:**
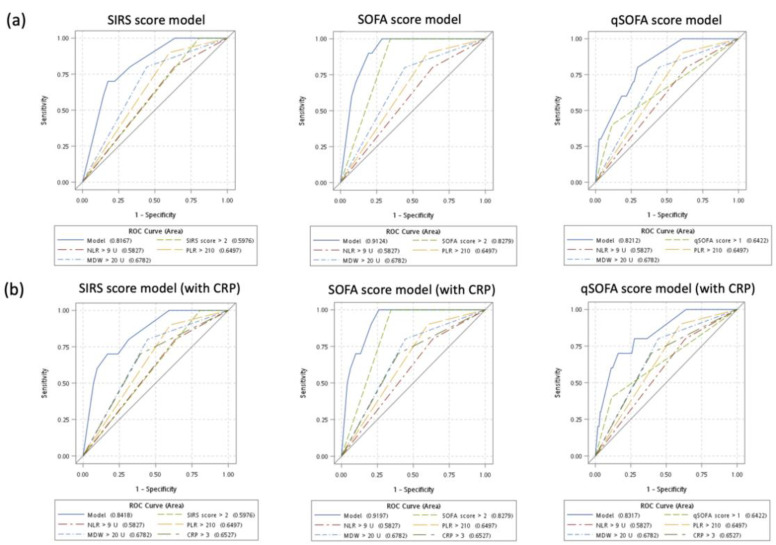
Receiver of operating characteristic (ROC) curves of predicting mortality for models based on SIRS score, SOFA score, and qSOFA score. (**a**) Models without CRP. (**b**) Models with CRP.

**Figure 4 jpm-11-00732-f004:**
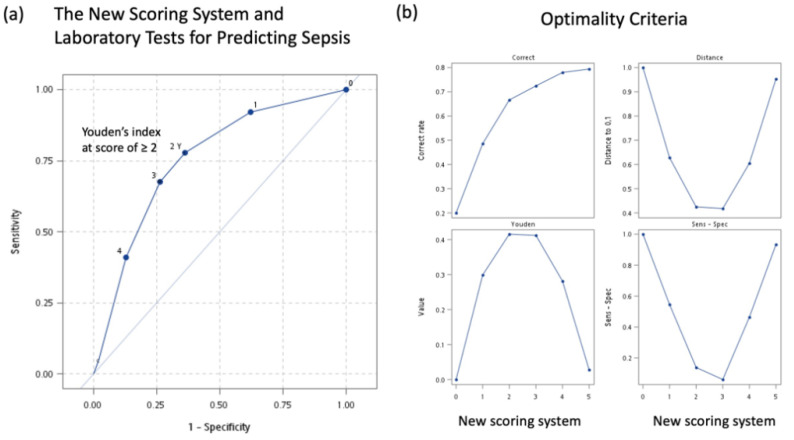
(**a**) Receiver of operating characteristic (ROC) curve of the New Scoring System and Laboratory Tests for Predicting Sepsis. (**b**) The optimality criteria of the New Scoring System and Laboratory Tests for Predicting Sepsis.

**Table 1 jpm-11-00732-t001:** Characteristics of Enrolled Patients.

	Before PS Matching (*n* = 8698)			After PS Matching (*n* =1480)		
Variables	Sepsis	Nonsepsis	SMD	Sepsis	Nonsepsis	SMD
Number (*n*)	308	8390		296	1184	
Age (years)	68.7 ± 17.4	58.5 ± 21.1	0.531	68.6 ± 17.7	69.4 ± 18.0	−0.040
Age subgroups *n*, (%)						
20–29 years	11 (3.6%)	938 (11.2%)		11 (3.7%)	39 (3.3%)	
30–39 years	12 (3.9%)	1074 (12.8%)		12 (4.1%)	51 (4.3%)	
40–49 years	22 (7.1%)	1055 (12.6%)		22 (7.4%)	91 (7.7%)	
50–59 years	40 (13.0%)	1103 (13.2%)		39 (13.2%)	141 (11.9%)	
60–69 years	52 (16.9%)	1334 (15.9%)		47 (15.9%)	183 (15.5%)	
70–79 years	87 (28.3%)	1191 (14.2%)		82 (27.7%)	276 (23.3%)	
≥80 years	84 (27.3%)	1695 (20.2%)		83 (28.0%)	403 (34.0%)	
Gender, *n* (%)			0.017			0.000
Female	165 (53.6%)	4470 (53.3%)		161 (54.4%)	644 (54.4%)	
Male	143 (47.4%)	3920 (46.7%)		135 (45.6%)	540 (45.6%)	
Glasgow coma scale (score)	13.5 ± 2.9	14.5 ± 1.8	−0.404	13.5 ± 2.9	13.7 ± 2.7	−0.082
Eye	3.7 ± 0.8	3.9 ± 0.4	−0.289	3.8 ± 0.7	3.8 ± 0.7	−0.072
Verbal	4.1 ± 1.5	4.7 ± 1.0	−0.410	4.2 ± 1.4	4.3 ± 1.3	−0.079
Motor	5.5 ± 1.1	5.8 ± 0.7	−0.366	5.5 ± 1.0	5.6 ± 1.0	−0.071
Triage levels (score)	2.2 ± 0.7	2.7 ± 0.6	−0.735	2.2 ± 0.7	2.2 ± 0.7	−0.015
Triage subgroups, *n* (%)						
Level 1	42 (13.6%)	366 (4.4%)		40 (13.5%)	157 (13.3%)	
Level 2	162 (52.6%)	2170 (25.9%)		152 (51.4%)	605 (51.1%)	
Level 3	104 (33.8%)	5843 (69.6%)		104 (35.1%)	420 (35.5%)	
Level 4	0 (0%)	11 (0.1%)		0 (0%)	2 (0.2%)	
Medical comorbidities, *n* (%)						
Hypertension	149 (48.4%)	2885 (34.4%)	0.293	143 (48.3%)	608 (51.4%)	−0.062
Diabetes mellitus	109 (35.9%)	1741 (20.8%)	0.345	106 (35.8%)	424 (35.3%)	0.000
Coronary artery disease	86 (27.9%)	1764 (21.0%)	0.152	81 (27.4%)	326 (27.5%)	−0.003
Cerebrovascular disease	26 (8.4%)	321 (3.8%)	0.198	25 (8.5%)	76 (6.4%)	0.085
End-stage renal disease	16 (5.2%)	209 (2.5%)	0.108	13 (4.4%)	59 (5.0%)	−0.033
Pulmonary disease	3 (1.0%)	59 (0.7%)	0.037	3 (1.0%)	16 (1.4%)	−0.037
Malignant disease	14 (4.6%)	292 (3.5%)	0.068	14 (4.7%)	53 (4.5%)	0.012

Continuous variables are expressed as the mean ± standard deviation. Abbreviations: *n*, number; PS, propensity score; SMD, standardized mean difference.

**Table 2 jpm-11-00732-t002:** Primary Outcome: The Scoring Systems and Laboratory Tests for Predicting Sepsis (Univariate).

Models	OR (95% CI)	*p*-Value	Area under Curve (95% CI)	Cutoff	Sensitivity (95% CI)	Specificity (95% CI)
Score systems (per score increase)						
SIRS score system	3.82 (3.24–4.56)	<0.0001 *	0.831 (0.812–0.849)			
SIRS score ≥ 3	5.46 (4.11–7.26)	<0.0001 *	0.660 (0.630–0.690)	3	44.9% (39.3–50.1%)	87.0% (85.1–88.9%)
SOFA score system	1.16 (1.11–1.22)	<0.0001 *	0.636 (0.604–0.668)			
SOFA score ≥ 3	1.90 (1.46–2.46)	<0.0001 *	0.576 (0.544–0.607)	3	47.0% (41.3–52.6%)	68.2% (65.6–70.8%)
qSOFA score system	1.49 (1.27–1.76)	<0.0001 *	0.580 (0.546–0.613)			
qSOFA score ≥ 2	1.86 (1.30–2.64)	0.0006 *	0.536 (0.513–0.560)	2	17.6% (13.2–21.9%)	89.7% (88.0–91.4%)
Laboratory tests						
WBC (per 10^3^/µL)	1.08 (1.06–1.10)	<0.0001 *	0.624 (0.585–0.664)			
WBC > 12	3.38 (2.59–4.52)	<0.0001 *	0.648 (0.585–0.664)	12	64.2% (58.7–69.7%)	65.4% (62.7–68.1%)
NLR (per U)	1.07 (1.05–1.08)	<0.0001 *	0.765 (0.735–0.795)			
NLR > 9	5.85 (4.43–7.73)	<0.0001 *	0.707 (0.678–0.737)	9	69.6% (69.4–74.8%)	71.9% (69.3–74.4%)
PLR (per 10 U)	1.02 (1.01–1.03)	<0.0001 *	0.666 (0.631–0.702)			
PLR > 160	2.61 (1.97–3.46)	<0.0001 *	0.610 (0.581–0.639)	160	73.3% (68.3–78.4%)	48.7% (45.9–51.6%)
PLR > 210	3.12 (2.39–4.05)	<0.0001 *	0.638 (0.607–0.669)	210	61.8% (56.3–67.4%)	65.8% (63.1–68.5%)
CRP (per mg/dL)	1.11 (1.09–1.13)	<0.0001 *	0.745 (0.721–0.779)			
CRP > 3	4.82 (3.66–6.36)	<0.0001 *	0.687 (0.658–0.717)	3	69.6% (64.4–74.8%)	67.8% (65.2–70.5%)
MDW (per U)	1.03 (1.01–1.05)	0.0056 *	0.639 (0.596–0.682)			
MDW > 20	2.77 (2.13–3.62)	<0.0001 *	0.625 (0.594–0.655)	22.1	64.5% (59.1–70.0%)	60.4% (57.6–63.2%)

Abbreviations: CRP, C-reactive protein; MDW, monocyte distribution width; NLR, neutrophil to lymphocyte ratio; OR, odds ratio; PLR, platelet-to-lymphocyte ratio; qSOFA, quick sequential organ failure assessment; SIRS, systemic inflammatory response syndrome; SOFA, sequential organ failure assessment; WBC, white blood cell count. * Statistical significance was defined at *p* < 0.05.

**Table 3 jpm-11-00732-t003:** Primary Outcome: the Scoring Systems and Laboratory Tests for Predicting Sepsis (Multivariate).

Characteristics	Multivariate Analysis(Model 1 on SIRS Score)		Multivariate Analysis(Model 2 on SOFA Score)		Multivariate Analysis(Model 3 on qSOFA Score)	
	OR (95% CI)	*p*-Value	OR (95% CI)	*p*-Value	OR (95% CI)	*p*-Value
**Multivariate Models without CRP**						
SIRS score ≥ 3	3.72 (2.73–5.06)	<0.0001 *				
SOFA score ≥ 3			1.57 (1.19–2.08)	0.0016 *		
qSOFA score ≥ 2					1.57 (1.07–2.30)	0.0210 *
NLR > 9 U	3.25 (2.31–4.58)	<0.0001 *	4.11 (2.94–5.74)	<0.0001 *	4.27 (3.08–5.97)	<0.0001 *
PLR > 210	1.48 (1.06–2.07)	0.0209 *	1.41 (1.01–1.95)	0.0413 *	1.36 (0.98–1.88)	0.0633
MDW > 20 U	1.90 (1.42–2.55)	<0.0001 *	1.95 (1.47–2.59)	<0.0001 *	1.98 (1.50–2.64)	<0.0001 *
Area under curve (95% CI)	0.796 (0.769–0.822)		0.761 (0.732–0.790)		0.757 (0.728–0.786)	
Hosmer-Lemeshow test	17.64 (9 groups)	0.0137 *	6.73 (8 groups)	0.3466	5.85 (7 groups)	0.3215
**Multivariate Models with CRP**						
SIRS score ≥ 3	3.45 (2.53–4.72)	<0.0001 *				
SOFA score ≥ 3			1.49 (1.12–1.99)	0.0065 *		
qSOFA score ≥ 2					1.46 (0.99–2.15)	0.0597
NLR > 9 U	2.88 (2.04–4.08)	<0.0001 *	3.53 (2.52–4.95)	<0.0001 *	3.66 (2.63–5.14)	<0.0001 *
PLR > 210	1.46 (1.04–2.04)	0.0274 *	1.39 (0.99–1.93)	0.0521	1.35 (0.97–1.87)	0.0744
MDW > 20 U	1.26 (0.91–1.74)	0.1622	1.27 (0.92–1.74)	0.1437	1.28 (0.93–1.75)	0.1281
CRP > 3 mg/dL	2.85 (2.06–3.94)	<0.0001 *	3.01 (2.20–4.15)	<0.0001 *	3.04 (2.22–4.19)	<0.0001 *
Area under curve (95% CI)	0.809 (0.784–0.835)		0.785 (0.757–0.813)		0.784 (0.757–0.812)	
Hosmer-Lemeshow test	16.18 (8 groups)	0.0128 *	5.43 (10 groups)	0.7109	5.60 (10 groups)	0.4692
Model comparison of IDI (%)	−1.93 (−3.33–−0.54)	0.0182 *	0.91 (−0.34–−2.17)	0.1924	0.70 (−0.57–−1.97)	0.3202

Abbreviations: CI, confidence interval; CRP, C-reactive protein; *n*, number; MDW, monocyte distribution width; NLR, neutrophil-to-lymphocyte ratio; OR, odds ratio; PLR, platelet-to-lymphocyte ratio; qSOFA, quick sequential organ failure assessment; SIRS, systemic inflammatory response syndrome; SOFA, sequential organ failure assessment. * Statistical significance was defined at *p* < 0.05.

**Table 4 jpm-11-00732-t004:** Secondary Outcome: the Scoring Systems and Laboratory for Mortality (Multivariate).

Characteristics	Multivariate Analysis(Model 1 on SIRS Score)		Multivariate Analysis(Model 2 on SOFA Score)		Multivariate Analysis(Model 3 on qSOFA Score)	
	OR (95% CI)	*p*-Value	OR (95% CI)	*p*-Value	OR (95% CI)	*p*-Value
**Multivariate Models without CRP**						
SIRS score ≥ 3	3.72 (2.73–5.06)	0.1051 †				
SOFA score ≥ 3			3.06 (1.46–∞)	0.0001 *,†		
qSOFA score ≥ 2					4.19 (1.15–15.35)	0.0304 *
NLR > 9 U	0.94 (0.17–5.23)	0.9446	0.49 (0.09–2.85)	<0.0001 *	0.58 (0.11–3.19)	0.5339
PLR > 210	0.15 (0.02–1.42)	0.0975	0.24 (0.02–2.40)	0.0413 *	0.18 (0.02–1.64)	0.1265
MDW > 20 U	7.16 (1.49–34.51)	0.0141 *	4.00 (0.83–19.32)	<0.0001 *	5.63 (1.16–27.40)	0.0325 *
Area under curve (95% CI)	0.817 (0.710–0.923)		0.912 (0.865–0.960)		0.821 (0.715–0.927)	
Hosmer-Lemeshow test	0.69 (7 groups)	0.9837	0.05 (4 groups)	0.9727	1.26 (7 groups)	0.9389
**Multivariate Models with CRP**						
SIRS score ≥ 3	−1.44 (−∞–0.21)	0.0802 *				
SOFA score ≥ 3			3.04 (1.44–∞)	0.0001 *		
qSOFA score ≥ 2					3.82 (1.03–14.11)	0.0445 *
NLR > 9 U	0.75 (0.13–4.17)	0.7402	0.41 (0.05–2.01)	0.3276	0.50 (0.09–2.71)	0.4192
LR > 210	0.15 (0.02–1.37)	0.0925	0.22 (0.01–1.62)	0.1996	0.17 (0.02–1.57)	0.1186
MDW > 20 U	4.23 (0.75–23.69)	0.1011	2.71 (0.58–19.66)	0.2450	3.69 (0.65–21.07)	0.1417
CRP >3	3.22 (0.70–14.78)	0.1326	2.65 (0.65–13.84)	0.1996	2.49 (0.53–11.68)	0.2469
Area under curve (95% CI)	0.842 (0.784–0.835)		0.920 (0.869–0.970)		0.832 (0.713–0.951)	
Hosmer-Lemeshow test	1.37 (7 groups)	0.9272	1.48 (4 groups)	0.4773	2.95 (8 groups)	0.4692

Abbreviations: CI, confidence interval; CRP, C-reactive protein; *n*, number; MDW, monocyte distribution width; NLR, neutrophil-to-lymphocyte ratio; OR, odds ratio; quick sequential organ failure assessment; PLR, platelet-to-lymphocyte ratio; qSOFA, quick sequential organ failure assessment; SIRS, systemic inflammatory response syndrome; SOFA, sequential organ failure assessment. * Statistical significance was defined at *p* < 0.05. † Exact logistic regression was used to estimate the OR.

**Table 5 jpm-11-00732-t005:** New Scoring Systems and Laboratory Tests for Predicting Sepsis.

	OR (95% CI)	*p* Value	Score	Sensitivity (95% CI)	Specificity (95% CI)
**Adopted model for building new scoring system**					
qSOFA score > 1	1.57 (1.07–2.30)	0.0210 *	1		
NLR > 9 U	4.27 (3.08–5.97)	<0.0001 *	2		
PLR > 210	1.36 (0.98–1.88)	0.0633	1		
MDW > 20 U	1.98 (1.50–2.64)	<0.0001 *	1		
Area under curve (95% CI)	0.757 (0.728–0.786)				
Hosmer–Lemeshow test	5.85 (7 groups)	0.3215			
**New scoring system**					
Score of ≥ 1	7.17 (4.61–11.15)	<0.0001 *		92.2% (89.2–95.3%)	37.7% (34.9–40.4%)
Score of ≥ 2	6.16 (4.57–8.30)	<0.0001 *		77.7% (73.0–82.4%)	63.9% (61.1–66.6%)
Score of ≥ 3	5.82 (4.42–7.67)	<0.0001 *		67.6% (62.2–72.9%)	73.7% (71.1–76.2%)
Score of ≥ 4	4.73 (3.55–6.31)	<0.0001 *		40.9% (35.3–46.5%)	87.3% (85.4–89.2%)
Score of ≥ 5	2.51 (1.27–4.93)	0.0078 *		4.8% (2.3–7.2%)	98.1% (97.3–98.8%)
Area under curve (95% CI)	0.755 (0.726–0.784)				
Hosmer–Lemeshow test	3.65 (5 groups)	0.3020			

Abbreviations: CI, confidence interval; MDW, monocyte distribution width; NLR, neutrophil-to-lymphocyte ratio; OR, odds ratio; PLR, platelet-to-lymphocyte ratio; qSOFA, quick sequential organ failure assessment. * Statistical significance was defined at *p* < 0.05.

## Data Availability

Data used in this study were electronic medical chart records and were not available in the public domain. Request of data should formally apply to Office of Human Research, Taipei Medical University, Taipei, Taiwan (tmujirb@gmail.com) and to the Joint Institutional Review Board, Taipei Medical University, Taipei, Taiwan.

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
