# Peer review of "Monocyte Distribution Width, Neutrophil-to-Lymphocyte Ratio, and Platelet-to-Lymphocyte Ratio Improves Early Prediction for Sepsis at the Emergency"

_jpm, 2021, doi:10.3390/jpm11080732_

Round 1
Reviewer 1 Report
Dear authors,
Thank you for giving me the opportunity to read and review your work. I think this is relevant research of a very relevant topic. However I do have some questions, and are very interested in your view of the comments I will give.
L 57: However, in real world clinical practice, these advanced biomarkers are not readily available in the ED.
- Although this might be true for IL-6 and PCT in certain settings, I really doubt this is the case for CRP. As to my knowledge this is a test even available as a point of care test in extramural settings. Would the authors consider CRP indeed more advanced than MDW?
L. 93: with infectious diseases,
- Is there anything known about the etiology (bacterial, virus, fungi) of the infectious diseases?
L. 101: The primary outcome was the early diagnosis of sepsis according to Sepsis-2 criteria.
- I have trouble with this being the sepsis definition. We know that around 10% of patients with sepsis will not fulfill the SIRS criteria (kaukonen et al. 2015 NEJM), on the other hand, many patients, will show SIRS features without ever developing a sepsis ('life threathening organ failure due/concurrent with an infection'). Although I can see why SIRS has an important place in the early identification of sepsis, as this is the population presenting at the ED with symptoms that are most commonly known to be associated with sepsis and previous research focused on early antibiotic treatment in this population, based international consensus they do not have sepsis. Could the authors explain the choice for SIRS?
L157. After applying the optimal cutoff point for sepsis scoring system 155
scores, a SIRS score of >2 (OR: 5.46, 95% CI: 4.11-7.26, P < .0001), a SOFA score of >2 (OR: 156
1.90, 95% CI: 1.46-2.46, P < .0001), and a qSOFA score of >1 (OR: 1.86, 95% CI: 1.30-2.64, P 157
< .0006) remained as significant predictors.
- If the independent variable is SIRS (from 0 to 4), and the dependent variable is basically SIRS (with a cut-off of 2 or more), isn't it strange to present this as a significant predictor? How are the patients with sepsis (2 or more SIRS criteria) any different from the patients with a SIRS score of 2, 3 and 4 combined? I do not think this is a valid multivariable analysis.
L158: Among these sepsis scoring systems, 44.9%, 158
47.0%, and 17.6% of patients exhibited low sensitivity for a SIRS score of >2, a SOFA score 159
of >2, and a qSOFA score of >1, respectively, and 87.0%, 68.2%, and 89.7% of patients 160
demonstrated high specificity, respectively.
- In my opinion, patients do no exhibit or show sensitivity or specificity, the tests are sensitive or specific for a certain disease.
In addition to the comment above: Among these sepsis scoring systems, 44.9%, 158
47.0%, and 17.6% of patients exhibited low sensitivity for a SIRS score of >2.
- Please explain this comment, what does this mean? How can only 44.9% of patients with sepsis have SIRS?
L. 161: In terms of laboratory tests, the biomarkers 161
WBC, NLR, PLR, CRP, and MDW were all found to be significant predictors.
Significant predictors of what? Sepsis? Would add to the sentence.
L.25 (discussion:' with and without sepsis'
Maybe reframe as with and without the need for early antibiotics, backboned by several references.
Although sepsis definition had changed, previous research on early antibiotic treatment, and resuscitation has commonly been performed in patients with SIRS.
L. 69 Discussion: ). This result suggested that the diagnostic performance of the new biomarkers was comparable to that of CRP.
- was there any added value, more specificity of the model? Could it be an earlier marker?
Discussion, general:
How would you describe the tool you report, as a tool to better identify patients who would benefit from antimicrobial therapy? from more intensive monitoring? From a more aggressive resuscitation?
Are the any statistics known on the antimicrobial etiology? were COVID-19 patients included?
Author Response
We thank the reviewers for their constructive comments. We have made revisions to the manuscript to address all the questions and comments raised by the three reviewers. We highlights changes made to the original version by setting the text color to red and “please see the attachment”. Our specific responses to each comment are as follows:
Responses to reviewers #1:
- Thank you for giving me the opportunity to read and review your work. I think this is relevant research of a very relevant topic. However I do have some questions, and are very interested in your view of the comments I will give.
Point 1: L 57: However, in real world clinical practice, these advanced biomarkers are not readily available in the ED. Although this might be true for IL-6 and PCT in certain settings, I really doubt this is the case for CRP. As to my knowledge this is a test even available as a point of care test in extramural settings. Would the authors consider CRP indeed more advanced than MDW?
- We are grateful for all of your constructive comments. We modified the following context in our manuscript.
- Introduction L59: In fact, interleukins, such as IL-6, are not readily available in our ED. PCT is an advanced biomarker and more specific for bacterial infection.1-3 However, PCT is not available in every first-level hospital, and its cost, which is 4 times higher than CRP, is also another concern. Although CRP is widely available in our hospital, the turnaround time (TAT) for CRP is longer than routine CBC report (including MDW)4-6.
- Two factors cause longer TAT for CRP in our ED. While the CRP is measured on clinical chemistry analyzer, the MDW is measured simultaneously with routine CBC on hematology analyzer. On average, the TAT for clinical chemistry analyzer is longer than the hematology analyzer.4-6 First, the principles of clinical chemistry analyzer is based on chemical reaction (with turbidimetry and nephelometry), and hematology analyzer is based on flow cytometry (with forward and side light scatter). Second, clinical chemistry analyzer has more extensive sorts of analytes than hematology analyzer. This causes prolonged time for sampling, dilution, and cleansing for reaction cells in clinical chemistry analyzer. In addition, Emergency physicians may thereafter order CRP based on the results of routine CBC analysis.
- Introduction L67: In our revised manuscript, we accentuated only “IL-6 and PCT” were not readily available biomarkers in ED. Under most condition, the turnaround time for clinical chemistry instrument is longer than TAT for hematology analyzer. If a point-of-care testing of CRP was not available, this may cause a delay of early antibiotic use.
- Point 2: 93: with infectious diseases, Is there anything known about the etiology (bacterial, virus, fungi) of the infectious diseases?
- In the revised manuscript, we addressed the identified infectious agents. In our ED, patients presenting SIRS routinely had one or more sets of blood culture taken. For patients with infectious diseases, a total of 397 sets of blood culture showed positive results. Of them, 250 (63.0%) sets of blood culture grew gram-negative bacilli, 134 (33.8%) sets grew gram-positive cocci, 11 sets (2.8%) grew gram-positive bacilli, and only 2 sets (0.5%) grew yeast. The detailed etiologies of microorganisms were summarized in the supplemental file (Table S1).
Supplemental Table S1. Etiologies of Microorganisms
|
Microorganisms |
Number |
Percentage (%) |
|
Gram-negative bacili |
250 |
63.0 |
|
Escherichia coli |
144 |
36.3 |
|
Klebsiella pneumoniae ssp pneumoniae |
27 |
6.8 |
|
Enterobacter cloacae complex |
10 |
2.5 |
|
Proteus mirabilis |
10 |
2.5 |
|
Pseudomonas aeruginosa |
10 |
2.5 |
|
Bacteroides fragilis |
6 |
1.5 |
|
Salmonella species |
6 |
1.5 |
|
Klebsiella oxytoca |
5 |
1.3 |
|
Aeromonas hydrophila |
4 |
1.0 |
|
Acinetobacter baumannii/calcoaceticus complex |
3 |
0.8 |
|
Citrobacter koseri |
3 |
0.8 |
|
Serratia marcescens |
3 |
0.8 |
|
Aeromonas caviae |
2 |
0.5 |
|
Campylobacter coli |
2 |
0.5 |
|
Enterobacter cloacae |
2 |
0.5 |
|
Providencia rettgeri |
2 |
0.5 |
|
Stenotrophomonas maltophilia |
2 |
0.5 |
|
Acinetobacter baumannii |
1 |
0.3 |
|
Bacteroides thetaiotaomicron |
1 |
0.3 |
|
Campylobacter jejuni |
1 |
0.3 |
|
Citrobacter freundii |
1 |
0.3 |
|
Klebsiella pneumoniae |
1 |
0.3 |
|
Klebsiella pneumoniae ssp ozaenae |
1 |
0.3 |
|
Leptotrichia goodfelloeii |
1 |
0.3 |
|
Morganella morganii |
1 |
0.3 |
|
Ralstonia pickettii |
1 |
0.3 |
|
Gram-positive cocci |
134 |
33.8 |
|
Staphylococcus aureus |
28 |
7.1 |
|
Staphylococcus capitis |
20 |
5.0 |
|
Staphylococcus epidermidis |
11 |
2.8 |
|
Staphylococcus hominis |
11 |
2.8 |
|
Streptococcus dysgalactiae |
8 |
2.0 |
|
Streptococcus agalactiae (Strep. group B) |
7 |
1.8 |
|
Enterococcus faecalis |
5 |
1.3 |
|
Streptococcus anginosus |
5 |
1.3 |
|
Staphylococcus haemolyticus |
4 |
1.0 |
|
Staphylococcus pettenkoferi |
4 |
1.0 |
|
Enterococcus faecium |
3 |
0.8 |
|
Enterococcus casseliflavus |
2 |
0.5 |
|
Lactococcus garvieae |
2 |
0.5 |
|
Micrococcus luteus |
2 |
0.5 |
|
Micromonas micros |
2 |
0.5 |
|
Staphylococcus lugdunensis |
2 |
0.5 |
|
Staphylococcus saprophyticus |
2 |
0.5 |
|
Streptococcus agalactiae |
2 |
0.5 |
|
Streptococcus gallolyticus |
2 |
0.5 |
|
Streptococcus pyogenes (Strep. group A) |
2 |
0.5 |
|
Aerococcus viridans |
1 |
0.3 |
|
Enterococcus hirae. |
1 |
0.3 |
|
Staphylococcus caprae |
1 |
0.3 |
|
Streptococcus constellatus |
1 |
0.3 |
|
Streptococcus cristatus |
1 |
0.3 |
|
Streptococcus dysgalactiae ssp equisimilis |
1 |
0.3 |
|
Streptococcus gordonii |
1 |
0.3 |
|
Streptococcus mitis |
1 |
0.3 |
|
Streptococcus oralis |
1 |
0.3 |
|
Streptococcus vestibularis |
1 |
0.3 |
|
Gram-positive bacilli |
11 |
2.8% |
|
Corynebacterium aurimucosum |
2 |
0.5 |
|
Lactobacillus rhamnosus |
2 |
0.5 |
|
Arcanobacterium haemolyticum |
1 |
0.25 |
|
Corynebacterium afermentans |
1 |
0.25 |
|
Corynebacterium argentoratense |
1 |
0.25 |
|
Corynebacterium minutissimum |
1 |
0.25 |
|
Corynebacterium striatum |
1 |
0.25 |
|
Erysipelothrix rhusiopathiae |
1 |
0.25 |
|
Solobacterium moorei |
1 |
0.25 |
|
Yeast |
2 |
2 |
|
Candida glabrata |
2 |
0.5 |
- Point 3: 101: The primary outcome was the early diagnosis of sepsis according to Sepsis-2 criteria. I have trouble with this being the sepsis definition. We know that around 10% of patients with sepsis will not fulfill the SIRS criteria (kaukonen et al. 2015 NEJM), on the other hand, many patients, will show SIRS features without ever developing a sepsis ('life threatening organ failure due/concurrent with an infection'). Although I can see why SIRS has an important place in the early identification of sepsis, as this is the population presenting at the ED with symptoms that are most commonly known to be associated with sepsis and previous research focused on early antibiotic treatment in this population, based international consensus they do not have sepsis. Could the authors explain the choice for SIRS?
- We concur a very small number of septic patients without overt SIRS symptoms, and some non-septic patients presenting SIRS. To our knowledge, the SIRS score as per Sepsis-2 criteria had better diagnostic performance for early identifying infectious diseases than the qSOFA or SOFA scores as per Sepsis-3 criteria did in the ED.7-12 The sepsis patients defined by SIRS comprised all qSOFA or SOFA patients in the ED11, whereas a number of sepsis patients defined by qSOFA or SOFA were not categorized into sepsis group by SIRS.
- The reason that sepsis-3 criteria replacing SIRS with qSOFA or SOFA is that some hemodynamic unstable non-infectious patients may also have high SIRS score.13, 14 Although the new Sepsis-3 criteria has much advantages of higher accuracy for predicting morbidity and mortality for patients in the intensive care unit.15, 16, replacing SIRS with newer Sepsis-3 definition in ED may cause an overlooked diagnosis of sepsis and may delay early antibiotic use.8
- For real-world practice, we make the diagnosis of sepsis by using SIRS score in ED. Since early antibiotic treatment is highly associated with a favorable outcome for septic patients, we did not allow any patients to be neglected or delayed for early antibiotic treatment in the ED. Our selection of SIRS as per Sepsis-2 criteria reflects the real practice in the ED. In addition, we summarize the SIRS score distribution between sepsis and non-sepsis groups in our supplemental file (Table S1).
Supplemental Table S1. SIRS Score Before and After Propensity Score Matching
|
|
Before PS matching (N = 8698) |
After PS matching (N =1480) |
||||
|
Variables |
Sepsis |
Nonsepsis |
P value |
Sepsis |
Nonsepsis |
P value |
|
Number (N) |
308 |
8390 |
|
296 |
1184 |
|
|
Mean SIRS score |
2.6 ± 0.7 |
1.1 ± 1.0 |
<0.001 |
2.5 ± 0.6 |
1.2 ± 1.0 |
<0.001 |
|
SIRS score ≥ 1 |
308 (100%) |
5528 (65.9%) |
<0.001 |
296 (100%) |
843 (71.2%) |
<0.001 |
|
SIRS score ≥ 2 |
308 (100%) |
2540 (30.3%) |
<0.001 |
296 (100%) |
452 (38.2%) |
<0.001 |
|
SIRS score ≥ 3 |
141(45.8%) |
691 (8.2%) |
<0.001 |
133 (44.9%) |
154 (13.0%) |
<0.001 |
|
SIRS score ≥ 4 |
27 (8.8%) |
61 (0.7%) |
<0.001 |
24 (8.1%) |
19 (1.6%) |
<0.001 |
- Point 4: After applying the optimal cutoff point for sepsis scoring system L155. scores, a SIRS score of >2 (OR: 5.46, 95% CI: 4.11-7.26, P < .0001), a SOFA score of >2 (OR: 156 1.90, 95% CI: 1.46-2.46, P < .0001), and a qSOFA score of >1 (OR: 1.86, 95% CI: 1.30-2.64, P 157 < .0006) remained as significant predictors. If the independent variable is SIRS (from 0 to 4), and the dependent variable is basically SIRS (with a cut-off of 2 or more), isn't it strange to present this as a significant predictor? How are the patients with sepsis (2 or more SIRS criteria) any different from the patients with a SIRS score of 2, 3 and 4 combined? I do not think this is a valid multivariable analysis.
- In our study, patients who were categorized into the “sepsis group” must fulfill the two inclusion criteria that was defined by Sepsis-2: (1) a SIRS score of ≥ 2 points, and (2) a documented infection source. Namely, the “sepsis group” in our study was not simply a SIRS score of ≥ 2 points. In addition, small number of patients in “non-sepsis” group had a SIRS score of ≥ 2 points. The SIRS score for patients between the sepsis and non-sepsis groups was added. (Please see the supplemental Table S2)
- Since our dependent variable is not simply a SIRS score of ≥ 2 but a combination of “SIRS score of ≥ 2” and “confirmed infection”, a SIRS score of >2 is reasonably to be used as an independent variable. To strengthen our explanation, we provide the data regarding the SIRS score distribution and its sensitivity and specificity at each cut-off value (Please see the supplemental Table S3).
Supplemental Table S2. SIRS Score Before and After Propensity Score Matching
|
|
Before PS matching (N = 8698) |
After PS matching (N =1480) |
||||
|
Variables |
Sepsis |
Nonsepsis |
P value |
Sepsis |
Nonsepsis |
P value |
|
Number (N) |
308 |
8390 |
|
296 |
1184 |
|
|
Mean SIRS score |
2.6 ± 0.7 |
1.1 ± 1.0 |
<0.001 |
2.5 ± 0.6 |
1.2 ± 1.0 |
<0.001 |
|
SIRS score ≥ 1 |
308 (100%) |
5528 (65.9%) |
<0.001 |
296 (100%) |
843 (71.2%) |
<0.001 |
|
SIRS score ≥ 2 |
308 (100%) |
2540 (30.3%) |
<0.001 |
296 (100%) |
452 (38.2%) |
<0.001 |
|
SIRS score ≥ 3 |
141(45.8%) |
691 (8.2%) |
<0.001 |
133 (44.9%) |
154 (13.0%) |
<0.001 |
|
SIRS score ≥ 4 |
27 (8.8%) |
61 (0.7%) |
<0.001 |
24 (8.1%) |
19 (1.6%) |
<0.001 |
Supplemental Table S3. Diagnostics for SIRS Score Before and After Propensity Score Matching
|
|
Before PS matching (N = 8698) |
After PS matching (N =1480) |
|||
|
Variables |
Sensitivity |
Specificity |
Sensitivity |
Specificity |
|
|
SIRS score ≥ 1 |
100.0% |
34.1% |
100.0% |
28.8% |
|
|
SIRS score ≥ 2 |
100.0% |
69.7 % |
100.0% |
61.8 % |
|
|
SIRS score ≥ 3 |
45.8% |
91.8% |
44.9% |
87.0% |
|
|
SIRS score ≥ 4 |
8.8% |
99.3% |
8.1% |
98.4% |
|
- Point5: L158: Among these sepsis scoring systems, 44.9%, 47.0%, and 17.6% of patients exhibited low sensitivity for a SIRS score of >2, a SOFA score of >2, and a qSOFA score of >1, respectively, and 87.0%, 68.2%, and 89.7% of patients demonstrated high specificity, respectively. In my opinion, patients do no exhibit or show sensitivity or specificity, the tests are sensitive or specific for a certain disease.
- Thank you for the comment. In our revised manuscript, we modified the section as follows: “Among these sepsis scoring systems, a SIRS score of ≥ 3, a SOFA score of ≥ 3, and a qSOFA score of ≥ 2, exhibited the low sensitivity of 9%, 47.0%, and 17.6%, respectively for sepsis, and the high specificity of 87.0%, 68.2%, and 89.7%, respectively, for sepsis”.
- Point 6: In addition to the comment above: Among these sepsis scoring systems, 44.9%, 47.0%, and 17.6% of patients exhibited low sensitivity for a SIRS score of >2.- Please explain this comment, what does this mean? How can only 44.9% of patients with sepsis have SIRS?
- Thank you for the comment. In our revised manuscript, we amended the symbol of “greater than (>)” to “greater than and equal (≥)” to improve readability. Therefore, we revised the description as “a SIRS score of ≥ 3 (previously a SIRS score of >2) exhibited the low sensitivity of 44.9% for sepsis”.
- In addition, the distribution of SIRS score, and its sensitivity and specificity were shown in supplemental files (Please see Table S2 and S3).
Supplemental Table S2. SIRS Score Before and After Propensity Score Matching
|
|
Before PS matching (N = 8698) |
After PS matching (N =1480) |
||||
|
Variables |
Sepsis |
Nonsepsis |
P value |
Sepsis |
Nonsepsis |
P value |
|
Number (N) |
308 |
8390 |
|
296 |
1184 |
|
|
Mean SIRS score |
2.6 ± 0.7 |
1.1 ± 1.0 |
<0.001 |
2.5 ± 0.6 |
1.2 ± 1.0 |
<0.001 |
|
SIRS score ≥ 1 |
308 (100%) |
5528 (65.9%) |
<0.001 |
296 (100%) |
843 (71.2%) |
<0.001 |
|
SIRS score ≥ 2 |
308 (100%) |
2540 (30.3%) |
<0.001 |
296 (100%) |
452 (38.2%) |
<0.001 |
|
SIRS score ≥ 3 |
141(45.8%) |
691 (8.2%) |
<0.001 |
133 (44.9%) |
154 (13.0%) |
<0.001 |
|
SIRS score ≥ 4 |
27 (8.8%) |
61 (0.7%) |
<0.001 |
24 (8.1%) |
19 (1.6%) |
<0.001 |
Supplemental Table S3. Diagnostics for SIRS Score Before and After Propensity Score Matching
|
|
Before PS matching (N = 8698) |
After PS matching (N =1480) |
|||
|
Variables |
Sensitivity |
Specificity |
Sensitivity |
Specificity |
|
|
SIRS score ≥ 1 |
100.0% |
34.1% |
100.0% |
28.8% |
|
|
SIRS score ≥ 2 |
100.0% |
69.7 % |
100.0% |
61.8 % |
|
|
SIRS score ≥ 3 |
45.8% |
91.8% |
44.9% |
87.0% |
|
|
SIRS score ≥ 4 |
8.8% |
99.3% |
8.1% |
98.4% |
|
- Point 7: L. 161: In terms of laboratory tests, the biomarkers WBC, NLR, PLR, CRP, and MDW were all found to be significant predictors. Significant predictors of what? Sepsis? Would add to the sentence.
- We added the description “the biomarkers WBC, NLR, PLR, CRP, and MDW were all found to be significant predictors of sepsis” in our revised manuscript.
- Point 8: L.25 (discussion:' with and without sepsis' Maybe reframe as with and without the need for early antibiotics, backboned by several references. Although sepsis definition had changed, previous research on early antibiotic treatment, and resuscitation has commonly been performed in patients with SIRS.
- In the revised manuscript, we reframed the discussion section as “with and without the need for early antibiotics.” The relevant discussion was added.
- Discussion L26: Since 2012, the early goal-directed therapy as per Sepsis-2 criteria had emphasized the benefit of antibiotic use in the first hour after severe sepsis or septic shock impressed.17 Every hour of delay in antibiotic administration was significantly associated with an increased mortality.18-20 Early use of proper antibiotics can be accomplished through prompt and correct diagnosis of sepsis.
- Point 9: L. 69 Discussion: ). This result suggested that the diagnostic performance of the new biomarkers was comparable to that of CRP. was there any added value, more specificity of the model? Could it be an earlier marker?
- In the revised manuscript, we addressed the additional values of new biomarkers in the discussion section. Recent studies and our analysis supported that MDW has comparable diagnostic accuracy like CRP for sepsis.21, 22 The sepsis scoring system in conjugation of MDW could increase the predicting performance of sepsis.
- Discussion L34: Firstly, Crouser et al. demonstrated early septic patients can be confirmed in the initial 12 hours of ED arrival with elevated MDW.23, 24 The time course of biomarkers after pathogen exposure in blood usually follows by (1) leukocytosis (such as count, morphological change of WBC), (2) inflammatory cytokines (such as IL-6, TNF-α), (3) inflammatory proteins (such as CRP, PCT), (4) poor perfusion-related markers (low pH, lactate), and (5) coagulopathy (elevated D-dimers, fibrin). On average, the peak concentration of PCT was around 18 hours and of CRP was around 24 to 48 hours. Compared to MDW, PCT and CRP are later biomarkers of sepsis in times course. Increased PCT and CRP giving an alarm to physicians that patients should be in a later phase of sepsis compared to increased MDW. The principle of MDW was based on morphological change of monocytes, which in conjugation with macrophages were important members of the mononuclear phagocyte system, the first line of innate immunity. To our knowledge, MDW should be used as an earlier marker and had comparable diagnostic performance of known sepsis biomarkers.
- Point 10: Discussion, general: How would you describe the tool you report, as a tool to better identify patients who would benefit from antimicrobial therapy? from more intensive monitoring? From a more aggressive resuscitation?
- In table 3, we proposed these 3 new biomarkers, including MDW, NLR, PLR, as a tool for better discrimination of sepsis patients from non-sepsis patients with similar characteristics and higher SIRS score. In table 4, our analysis supported an increase of MDW >20 is associated with early mortality within 3 days due to septic shock.
- The developed 5-point tool in Table 5 supported a simplified algorithm for sepsis management in ED, where a patient with new score ≥ 2 (any combination of the sepsis symptoms and elevated biomarkers of MDW >20, NLR >9, and MDW >20) should immediately receive antibiotic to decrease the progression of sepsis diseases.
- MDW is reported in conjugation with routine CBC, and it could be easily used for more frequent and intensive monitoring in the ward and intensive care units. Since this study was conducted primarily in ED, all patients were followed up for 3 days. Further studies are warranted to assess whether intensive monitoring of biomarkers for critical ill patients helps improve the long-term outcome.
- Point 11: Are the any statistics known on the antimicrobial etiology? were COVID-19 patients included?
- Data regarding the antimicrobial etiology was not available in our study. However, in our clinical practice, the first line broad-spectrum and site-specific antibiotics were administrated once sepsis was diagnosed in ED. De-escalation antibiotic therapy was considered afterwards according to results of antimicrobial etiology and susceptibility, and the clinical condition.
- Result L 151: During the study period from January 01, 2020, and November 30, 2020, the COVID-19 was not a pandemic disease in Taiwan. None of the patients enrolled in the study had the diagnosis of COVID-19. According to data from the Taiwan National Infectious Disease Statistics System (please see the figure below), no indigenous case during our study period was reported.
The figure was adapted from the website of Ministry of Health and Welfare, Taiwan on July 24 (https://nidss.cdc.gov.tw/en/nndss/disease?id=19CoV).
References
- Pierce R, Bigham MT, Giuliano Jr JS. Use of procalcitonin for the prediction and treatment of acute bacterial infection in children. Current Opinion in Pediatrics. 2014;26(3):292-298.
- Schuetz P, Albrich W, Christ-Crain M, Chastre J, Mueller B. Procalcitonin for guidance of antibiotic therapy. Expert review of anti-infective therapy. 2010;8(5):575-587.
- Rey C, Los Arcos M, Concha A, et al. Procalcitonin and C-reactive protein as markers of systemic inflammatory response syndrome severity in critically ill children. Intensive care medicine. 2007;33(3):477-484.
- Hawkins RC. Laboratory turnaround time. The Clinical Biochemist Reviews. 2007;28(4):179.
- Lee EJ, Do Shin S, Song KJ, et al. A point-of-care chemistry test for reduction of turnaround and clinical decision time. The American journal of emergency medicine. 2011;29(5):489-495.
- Kankaanpää M, Raitakari M, Muukkonen L, et al. Use of point-of-care testing and early assessment model reduces length of stay for ambulatory patients in an emergency department. Scandinavian journal of trauma, resuscitation and emergency medicine. 2016;24(1):1-7.
- Williams JM, Greenslade JH, McKenzie JV, Chu K, Brown AF, Lipman J. Systemic inflammatory response syndrome, quick sequential organ function assessment, and organ dysfunction: insights from a prospective database of ED patients with infection. Chest. 2017;151(3):586-596.
- Prasad PA, Fang MC, Abe-Jones Y, Calfee CS, Matthay MA, Kangelaris KN. Time to Recognition of Sepsis in the Emergency Department Using Electronic Health Record Data: A Comparative Analysis of SIRS, SOFA, and qSOFA. Critical care medicine. 2020;48(2):200.
- Graham CA, Leung LY, Lo RSL, Yeung CY, Chan SY, Hung KKC. NEWS and qSIRS superior to qSOFA in the prediction of 30-day mortality in emergency department patients in Hong Kong. Annals of Medicine. 2020;52(7):403-412.
- Mak MHW, Low JK, Junnarkar SP, Huey TCW, Shelat VG. A prospective validation of Sepsis-3 guidelines in acute hepatobiliary sepsis: qSOFA lacks sensitivity and SIRS criteria lacks specificity (Cohort Study). International Journal of Surgery. 2019;72:71-77.
- Luo J, Jiang W, Weng L, et al. Usefulness of qSOFA and SIRS scores for detection of incipient sepsis in general ward patients: A prospective cohort study. Journal of critical care. 2019;51:13-18.
- Gando S, Shiraishi A, Abe T, et al. The SIRS criteria have better performance for predicting infection than qSOFA scores in the emergency department. Scientific reports. 2020;10(1):1-9.
- Chakraborty RK, Burns B. Systemic inflammatory response syndrome. 2019;
- Cinel I, Opal SM. Molecular biology of inflammation and sepsis: a primer. Critical care medicine. 2009;37(1):291-304.
- Poutsiaka DD, Porto MC, Perry WA, et al. Prospective observational study comparing Sepsis-2 and Sepsis-3 definitions in predicting mortality in critically ill patients. Oxford University Press US; 2019:ofz271.
- Khwannimit B, Bhurayanontachai R, Vattanavanit V. Comparison of the performance of SOFA, qSOFA and SIRS for predicting mortality and organ failure among sepsis patients admitted to the intensive care unit in a middle-income country. Journal of critical care. 2018;44:156-160.
- Dellinger RP, Levy MM, Rhodes A, et al. Surviving Sepsis Campaign: international guidelines for management of severe sepsis and septic shock, 2012. Intensive care medicine. 2013;39(2):165-228.
- Ferrer R, Artigas A, Suarez D, et al. Effectiveness of treatments for severe sepsis: a prospective, multicenter, observational study. American journal of respiratory and critical care medicine. 2009;180(9):861-866.
- Barie PS, Hydo LJ, Shou J, Larone DH, Eachempati SR. Influence of antibiotic therapy on mortality of critical surgical illness caused or complicated by infection. Surgical infections. 2005;6(1):41-54.
- Castellanos-Ortega Á, Suberviola B, García-Astudillo LA, et al. Impact of the Surviving Sepsis Campaign protocols on hospital length of stay and mortality in septic shock patients: results of a three-year follow-up quasi-experimental study. Critical care medicine. 2010;38(4):1036-1043.
- Hausfater P, Boter NR, Indiano CM, et al. Monocyte distribution width (MDW) performance as an early sepsis indicator in the emergency department: comparison with CRP and procalcitonin in a multicenter international European prospective study. Critical Care. 2021;25(1):1-12.
- Woo Al, Oh DK, Park C-J, Hong S-B. Monocyte distribution width compared with C-reactive protein and procalcitonin for early sepsis detection in the emergency department. PloS one. 2021;16(4):e0250101.
- Crouser ED, Parrillo JE, Seymour CW, et al. Monocyte distribution width: a novel indicator of sepsis-2 and sepsis-3 in high-risk emergency department patients. Critical care medicine. 2019;47(8):1018.
- Crouser ED, Parrillo JE, Seymour C, et al. Improved early detection of sepsis in the ED with a novel monocyte distribution width biomarker. Chest. 2017;152(3):518-526.

Reviewer 2 Report
The manuscript proposes the new system for sepsis prediction in early stage. The strenghts of the study is a big number of patients and the simplicity of the proposed system, which does not require more staff as well as technical equipement, beside the quite advanced hematology analyzer. The weaknesses are some of the references - specifically reference 18 and 19 do not mention "monocyte distribution width". They should be removed and replaced by relevent references. Also there is missing an important reference by Djordjevic et al. (PMID: 30116146). Otherwise, the manuscript is clear and shows the useful data that are important for ED.
Author Response
We thank the reviewers for their constructive comments. We have made revisions to the manuscript to address all the questions and comments raised by the three reviewers. We highlights changes made to the original version by setting the text color to red and "please see the attachment". Our specific responses to each comment are as follows:
Responses to reviewers #1:
- Point: The manuscript proposes the new system for sepsis prediction in early stage. The strengths of the study is a big number of patients and the simplicity of the proposed system, which does not require more staff as well as technical equipment, beside the quite advanced hematology analyzer. The weaknesses are some of the references - specifically reference 18 and 19 do not mention "monocyte distribution width". They should be removed and replaced by relevent references. Also there is missing an important reference by Djordjevic et al. (PMID: 30116146). Otherwise, the manuscript is clear and shows the useful data that are important for ED.
- We are grateful for all of your constructive comments. In the revised manuscript, we addressed all of the additional points in the following sections. We replaced the improper reference 18 and 19 with an important reference by Djordjevic et al. (PMID: 30116146) and others, and we revised the text in the manuscript.
- L66-68: Monocyte distribution width (MDW), a new biomarker analyzed in conjugation with routine CBC on hematology analyzer, was found to elevate in the ongoing sepsis.9, 15, 18
- We amended the text with an important reference by Djordjevic et al. (PMID: 30116146) as follows.
- Diordjevic et al. even noticed that level of NLR and PLR correlated to the result of blood culture.23 (Discussion section, Line 55-56)
- In addition, other biomarkers examined using a hematology analyzer including the neutrophil-to-lymphocyte ratio (NLR)21-23 and platelet-to-lymphocyte ratio (PLR)23-25 are useful for the early diagnosis of sepsis. (Introduction section, Line 71-74)
- Besides, previous studies have found that NLR and PLR were significantly associated with the prediction and prognosis of sepsis.22, 37-40 (Discussion section, Line 53-55)
- Our cutoff values are consistent with the reference intervals of NLR (ranging from 4.06 to 23.80)23, 41-46 and PLR (ranging from 200 to 240)23, 24, 45 indicated in previous studies. (Discussion section, Line 58-60)
Reference:
Djordjevic D, Rondovic G, Surbatovic M, et al. Neutrophil-to-lymphocyte ratio, monocyte-to-lymphocyte ratio, platelet-to-lymphocyte ratio, and mean platelet volume-to-platelet count ratio as biomarkers in critically ill and injured patients: which ratio to choose to predict outcome and nature of bacteremia? Mediators of inflammation. 2018;2018
